# Click Beetle Mitogenomics with the Definition of a New Subfamily Hapatesinae from Australasia (Coleoptera: Elateridae)

**DOI:** 10.3390/insects12010017

**Published:** 2020-12-29

**Authors:** Dominik Kusy, Michal Motyka, Ladislav Bocak

**Affiliations:** Laboratory of Biodiversity and Molecular Evolution, CATRIN-CRH, Palacky University, 17. listopadu 50, 771 46 Olomouc, Czech Republic; dominik.kusy@upol.cz (D.K.); michal.motyka@upol.cz (M.M.)

**Keywords:** taxonomy, mitochondrial genomes, phylogeny, Gondwana, Australian region, new subfamily, new status

## Abstract

**Simple Summary:**

The classification of click beetles is revisited with newly sequenced mitochondrial genomes of eight species belonging to seven different subfamilies. The genus *Hapatesus* Candèze, 1863, is herein excluded from Dentrometrinae and designated as a type genus of Hapatesinae, a new subfamily. Phylogenetic analyses suggest that Eudicronychinae is a terminal lineage in Elaterinae. Consequently, we propose Eudicronychini, a new status. The deep mtDNA-based split between Elaterinae and the clade of other click beetle subfamilies agrees with the results of phylogenomic analyses and mitochondrial genomes provide a sufficient signal for inference of shallow splits.

**Abstract:**

Elateridae is a taxon with very unstable classification and a number of conflicting phylogenetic hypotheses have been based on morphology and molecular data. We assembled eight complete mitogenomes for seven elaterid subfamilies and merged these taxa with an additional 22 elaterids and an outgroup. The structure of the newly produced mitogenomes showed a very similar arrangement with regard to all earlier published mitogenomes for the Elateridae. The maximum likelihood and Bayesian analyses indicated that *Hapatesus* Candèze, 1863, is a sister of Parablacinae and Pityobiinae. Therefore, Hapatesinae, a new subfamily, is proposed for the Australian genera *Hapatesus* (21 spp.) and *Toorongus* Neboiss, 1957 (4 spp.). Parablacinae, Pityobiinae, and Hapatesinae have a putative Gondwanan origin as the constituent genera are known from the Australian region (9 genera) and Neotropical region (*Tibionema* Solier, 1851), and only *Pityobius* LeConte, 1853, occurs in the Nearctic region. Another putative Gondwanan lineage, the Afrotropical Morostomatinae, forms either a serial paraphylum with the clade of Parablacinae, Pityobiinae, and Hapatesinae or is rooted in a more terminal position, but always as an independent lineage. An Eudicronychinae lineage was either recovered as a sister to Melanotini or as a deep split inside Elaterinae and we herein transfer the group to Elaterinae as Eudicronychini, a new status. The mitochondrial genomes provide a sufficient signal for the placement of most lineages, but the deep bipartitions need to be compared with phylogenomic analyses.

## 1. Introduction

Click beetles, Elateridae, are a major elateroid family with ~10,000 recognized species from all zoogeographic regions. The group is well-known even to non-specialists due to its common occurrence in all ecosystems and characteristic clicking mechanism [1] (Figure 1A–D). The clicking elateroids are morphologically very uniform and they resemble the general appearance of false click beetles (Eucnemidae; Figure 1E) and small false click beetles (Throscidae) which are distantly related [2]. The characteristic slender body shape, rectangular to acutely projected posterior angles of the pronotum, and short slender legs and antennae, both often partly sunk in grooves in a resting position, are characteristic of most elaterids (Figure 1A–D). A dependence on clicking as an antipredator strategy possibly constrains the morphological evolution of the group because the mechanism depends on precise interaction between pro- and mesothoracic structures that keep the body in a brace position until their release [3,4]. Some forms similar to extant taxa have been reported already from the Late Triassic and Early Jurassic, the early-Cretaceous Jinju formation, and Burmese, Dominican, and Baltic amber inclusions [5,6,7,8].

Despite high morphological uniformity (Figure 1), the earlier students of click beetles described a number of family group taxa, i.e., subfamilies, tribes, and subtribes (altogether >130 taxa; [9]), but their relationships have remained contentious, reliable diagnostic characters are absent, constituent genera are often transferred between subfamilies or tribes, and the ranks of individual taxa are unstable [1,10]. Not only morphology-based hypotheses on relationships, but also Sanger molecular data have provided weak support for the click beetle classification, and additionally, recovered topologies were in conflict with previous definitions of separate “cantharoid” and “elateroid” groups of families [11]. The morphological delimitation of click beetles has been recently seriously questioned by the inclusion of some soft-bodied, morphologically highly divergent “cantharoid” lineages into Elateridae. Already, Bocakova et al. [2] showed that *Drilus* Olivier, 1790, believed at the time to be a genus of the family Drilidae *sensu* Crowson [12], might be a modified click beetle and its position in the Agrypninae was confirmed with further mito-ribosomal data [13,14]. Later, Plastoceridae were transferred to Elateridae and given the rank of a subfamily [15]. Due to limited statistical support of mito-ribosomal phylogenies for deep relationships [2,13,14,15,16], the results were considered by most students of click beetles with reluctance [17,18], see [19] for further information. Therefore, the Sanger data for click beetles have not recently been intensively produced, and only a few sequences for elaterid taxa have been reported in the last years to identify relationships of some species-poor lineages, e.g., Pityobiinae and Parablacinae [20]. Similarly, a limited number of protein-coding nuclear genes did not perform well, and click beetles were not identified as a monophylum when a large dataset for the whole Coleoptera was analyzed [21].

Next-generation sequencing opened a possibility to assemble large datasets for phylogenetic analyses. The early phylogenomic studies aimed at deep relationships in Coleoptera and, therefore, click beetles were represented by a few terminals and the family Elateridae was sometimes not recovered as a monophylum in contrast with its morphological uniformity [22,23,24]. Nevertheless, the studies consensually showed that two main lineages can be delimited: the Elaterinae and the clade containing other click beetle subfamilies (Dendrometrinae, Agrypninae, Cardiophorinae, Negastriinae, etc.). Later studies using transcriptomic data and whole-genome shotgun sequencing robustly supported earlier inclusion of Drilini (in Agrypninae), Plastocerinae, and Omalisinae within Elateridae [25]. Additionally, genomic data indicated the close relationships of clicking and non-clicking elateroids, and a family Sinopyrophoridae was proposed for an elaterid-like sister group of bioluminescent, soft-bodied fireflies and glowworm beetles (Lampyridae, Phengodidae, Rhagophthalmidae [26]). Simultaneously, these analyses have shown that further data are needed to propose a robust phylogeny that can serve as a basis for the natural classification of click beetles. 

After the delimitation of the non-Elaterinae clade of click beetles, we should more intensively study the relationships between constituent lineages. Herein, we report newly assembled mitochondrial genomes. With these data, we investigate the relationships between several Gondwanan click beetle lineages, i.e., Parablacinae and Pityobiinae, and another lineage of Australian elaterids represented in our study by *Hapatesus* Candèze, 1863. The Australian fauna evolved in the last 100 million years in relative isolation and houses a number of ancient lineages in various insect groups [27]. As click beetles have been reported from the oldest deposits containing recognizable extant beetle families [5,6,7], the Australian fauna can be a museum of ancient diversity and deserves close attention. 

## 2. Materials and Methods

*Materials.* The newly sequenced samples (Table 1) were preserved in the field in ethanol and kept at −20 °C until extraction of DNA, or a dry mounted specimen was used. The voucher specimens were used for morphological investigation. They were dissected after short relaxation in 50% ethanol. Fat and muscles were removed by keeping the dissected body parts in hot 10% KOH for a short time and some structures were cleared using a hot aqueous solution of hydrogen peroxide and lactic acid. The photographs were taken by a Canon EOS D700 camera attached to an Olympus SZX16 binocular microscope. Stacks of photographs were assembled using Helicon Focus software and processed in Photoshop 6.0. All vouchers of newly sequenced specimens are deposited at the voucher collection of the Laboratory of Biodiversity & Molecular Evolution at Czech Advanced Technology and Research Institute (CATRIN), Palacky University.

*Laboratory procedures and data handling.* Genomic DNA was extracted from metathoracic muscles using MagAttract HMW DNA extraction kit and eluted in 150 μL of AE buffer and kept at −80 °C until sequencing. Short insert size library constructions (~320 bp) and subsequent paired-end (2 × 150 bp) sequencing of the samples were done by Novogene, Inc., Beijing using Illumina Hiseq Xten. We used raw Illumina reads from an earlier study [25] for an assembly of two mitogenomes (*Drilus mauritanicus* and *Plastocerus angulosus*; Table 1). 

Raw Illumina reads were quality checked with FastQC and filtered with fastp 0.21.0 [28] using -q 28 -u 50 -n 15 -l 50 settings. Filtered reads were used for final mitogenome assemblies. The mitogenomes were built de novo using the NOVOPlasty v.2.7.2 pipeline [29]. NOVOPlasty was run with the default settings except the kmer value when we used a multi kmer strategy with the following kmer sizes of 25, 39, 45, and 51. We used as seed the single fragment of *Oxynopterus* sp. *cox1* gene available in GenBank (HQ333982). The newly assembled mitochondrial genomes were annotated using the MITOS2 webserver [30] with the invertebrate genetic code and RefSeq 63 metazoa reference. Software ARWEN [31] implemented in the MITOS2 web server was used for the identification of tRNA genes. The annotation, circularization, and start + stop codons corrections of protein-coding genes (PCSGs) were performed manually in Geneious 7.1.9. The visualization of genomes was conducted with OrganellarGenomeDRAW [32]. The sequences of newly produced mitochondrial genomes were submitted to GenBank (Table 1).

*Phylogenetic analyses.* The eight newly assembled and annotated mitochondrial genomes of click beetles were merged for the purpose of phylogenetic analyses with additional 23 highly complete mitochondrial genomes (22 ingroup taxa and 1 outgroup) available at the GenBank database. Most of them were reported in earlier studies [33,34,35,36,37] and some were released from the Darwin Initiative (Natural History Museum, London) without prior analysis. In total, the phylogenetic analyses included 30 terminals belonging to ten subfamilies of Elateridae. The list of earlier published mitogenomes used for phylogenetic analyses is available in Appendix A.

The sequences of 13 PCGs and two rRNA genes were extracted from analyzed mitogenomes. The nucleotide sequences of PCGs were aligned using TransAlign [38]. In addition, nucleotide sequences of rRNA genes and translated amino acid sequences of PCGs were aligned with Mafft v.7.407 using the L-INS-i algorithm [39]. The aligned data were concatenated with FASconCAT-G v.1.04 [40]. We compiled the following datasets: (A) NUC123: 15 mitochondrial genes were partitioned by gene and PCGs were further partitioned by codon position; (B) PCN12: 13 mitochondrial PCGs partitioned by gene and by first and second codon positions with third codon position removed; (C) AA: amino acids of 13 mitochondrial PCGs partitioned by gene; (D) MTallAS: dataset A analysed using AliScore to remove possibly ambiguously aligned regions.

The degree of missing data and overall pairwise completeness scores across all datasets was inspected using AliStat v.1.7. AliGROOVE [41] was used to analyse sequence divergence heterogeneity with the default sliding window size. Indels in the nucleotide datasets were treated as ambiguities and a BLOSUM62 matrix was used as the default amino acid substitution matrix. The software SymTest v.2.0.49 with Bowker’s test was used to calculate the deviation from stationarity, reversibility, and homogeneity [42] (SRH). Heatmaps were generated for all datasets to visualize the pairwise deviations from SRH conditions.

Phylogenetic inferences were performed under maximum likelihood (ML) optimization using IQ-Tree2.1.2 [43], and Bayesian inference (BI) using PhyloBayes MPI v.1.8 [44]. Before ML tree searches, best-fitting model selection for each partition was performed with ModelFinder [45,46] using the -MFP and -merit BIC options. For the amino acid dataset, the substitution models mtART, mtZOA, mtMet, mtInv, Blosum62LG, DCMUT, JTT, JTTDCMUT, DAYHOFF, WAG, and free rate models LG4X and LG4M were considered. The nucleotide datasets were tested against a complete list of models. Further, all combinations of rate heterogeneity among sites were allowed (options: -mrate E,I,G,I + G,R-gmedian). We used the edge-linked partitioned model for tree reconstructions (-spp option) allowing each partition to have its own rate. Ultrafast bootstrap [47], SH-like approximate likelihood ratio test (SH-aLRT), and aBayes test were calculated for each tree using -bb 5000 -alrt 5000 and -abayes options. To avoid situations when ML search falls to local maxima, we replicated the analysis using 50 independent runs with random starting trees and 50 runs with 100 parsimonious and BioNJ starting trees as the default setting in IQ-TREE and tree likelihood scores were compared.

In the PhyloBayes analysis, unpartitioned datasets A, B, and C were analyzed under the site-heterogeneous mixture CAT + GTR + Γ4 model for all searches. Two independent Markov chain Monte Carlo (MCMC) were run for each dataset. We checked for the convergence in the tree space with bpcomp program and generated output of the largest (maxdiff) and mean (meandiff) discrepancy observed across all bipartitions and generated a majority-rule consensus tree using a burn-in of 30% and sub-sampling every 10th tree. Additionally, we used the program tracecomp to check for convergence of the continuous parameters of the model.

We applied further tests of alternative phylogenetic relationships using the approximately unbiased AU-test [48], p-SH: *p*-value of the Shimodaira–Hasegawa test [49]; KH-test: one-sided Kishino–Hasegawa test [50]; p-WKH: *p*-value of weighted KH test; p-WSH: *p*-value of weighted SH test, c-ELW: Expected Likelihood Weight (ELW) [51]; bp-RELL: bootstrap proportion using RELL method [52]. Calder [53] listed the genus *Hapatesus* in the Dendrometrinae subfamily. Therefore, the result of ML tree search was tested against alternative topologies where *Hapatesus* sp. was placed (A) in a sister position to the clade containing Dentrometrinae, Morostomatinane, Cardiophorinae, and Agrypninae taxa; (B) in a sister position to the Dendrometrinae clade; and (C) in a terminal position inside Dendrometrinae. All tests were performed in IQ-TREE2 [43] testing per site log-likelihoods using the -zb 50,000 -zw -au parameters.

## 3. Results

### 3.1. Structure of Mitochondrial Genomes

All eight newly sequenced and assembled mitogenomes contained the entire set of 37 genes usually present in insect mitogenomes (13 PCGs, 22 tRNA genes, and two rRNA genes), and a large non-coding region (control region). The gene order of the newly sequenced Elateridae species followed the presumed ancestral arrangement of beetle mitogenomes (Appendix A, Figure 2 and Appendix A). The length of complete mitogenomes ranged from 15.9 (*Eudicronychus rufus*) to 17.8 kbp (*Plastocerus angulosus*). For detailed information about gene positions, direction, and overall gene order see Appendix A and Appendix A.

### 3.2. Analyses of Compositional Heterogeneity, Sequence Heterogeneity, and Sequence Completeness

Symtest analyses showed a high compositional heterogeneity among click-beetle mitogenomes (the dataset A, B, and D, but not the dataset C; Figure 2B; Appendix A) with the following percentages of pairwise *p*-values < 0.05 rejecting SRH conditions in analyzed datasets: (A) *p*-values < 0.05: 85.38%, (B) 58.49%, (C) 8.82%, and (D) 84.30%. Additionally, heterogeneity of sequence variation was assessed with AliGROOVE, separately for all datasets (Appendix A). In general, the mitogenomes had low heterogeneity of sequence composition for most pairwise comparisons between the sequences. Only sequences of *Agriotes ustulatus*, *Dicronychus* sp., *Drilus flavescens*, and *Pyrophorus divergens* showed higher levels of sequence heterogeneity across some datasets, but it is most likely caused by missing data. Conversely, the 3rd codon positions of 13 PCGs showed very high levels of heterogeneity. All the datasets show overall a high level of pairwise completeness. AliStat completeness score for the alignment (Ca) was 0.95–0.97. (Appendix A for details).

### 3.3. Phylogenetic Relationships

The Bayesian and maximum likelihood analyses of complete mitogenomes used partitions as shown in Appendix A and the analyses produced well-resolved topologies with high support for tribe-level relationships, but lower support for deep bipartitions (Figure 3 and Appendix A). Two major clades were recovered, i.e., Elaterinae and the remaining subfamilies. The subfamilies constituting the second clade included Dendrometrinae, Cardiophorinae, and Agrypninae and three subfamilies which predominantly occur in Gondwanan continents, i.e., Pityobiinae, Parablacinae, and *Hapatesus*. These three lineages were found as a monophylum. *Hapatesus*, currently placed in Dendrometrinae, was sister to the clade of Parablacinae and Pityobiinae. Although the support for its relationship was variable in ML analyses of the dataset PCN12 (ALRT/UFBoot 98/75), the Bayesian analysis supported its position with high posterior probability PP = 1.0, and the analysis of the NUC123 dataset recovered the relationships with statistical support (ALRT/UFBoot 99/78, PP 1.0). Six analyses found *Hapatesus* to be sister of Pityobiinae and Parablacinae, all of them supporting the relationship with either ALRT ≥ 97% or PP = 1.0 (two analyses). A single Bayesian analysis (amino acid dataset) recovered PP = 0.79 for the same position of *Hapatesus*. Although the lower support values were recovered in some analyses, the position of *Hapatesus* as a sister to Pityobiinae and Parablacinae was stable in all recovered topologies. The ML analysis of the AA dataset recovered *Hapatesus* as a sister to Morostomatinae (Figure 3 and Appendix A). Even this incongruent position does not support relationship of *Hapatesus* and Dendrometrinae, it does not place the taxon within any earlier defined subfamily of click beetles and justifies the separate subfamily-rank position of *Hapatesus*. Another Gondwanan subfamily, Morostomatinae, represented by *Diplophoenicus*, was found as sister to three crown subfamilies, i.e., Dendrometrinae, Cardiophorinae, and Agrypninae, but the position of Morostomatinae obtained variable support (Figure 3 and Appendix A). *Eudicronychus* Méquignon, 1931, the type genus of Eudicronychinae, represents the terminal split in Elaterinae and was recovered as a sister to *Melanotus* Eschscholtz, 1829. Its position is firmly held by the robust support for deep bipartitions among tribes of Elaterinae in all analyses (Figure 3A,B and Appendix A).

We have tested several approaches in the analysis of the dataset and we found that, although some variability is present in some bipartitions, all analyses confirm the position of both focal clades, i.e., *Hapatesus* as a representative of a separate deeply rooted lineage with the highest affinity to the Australian clade of click beetles. All tests reject the earlier position of *Hapatesus* in Dendrometrinae (Table 2). Additionally, two species of *Eudicronychus* Méquignon, 1931, closely related to *Eudicronychus*, the type genus of the earlier recognized subfamily Eudicronychinae or the family Eudicronychidae, were recovered as a lineage within Elaterinae (Figure 3 and Appendix A). Most analyses indicate that *Plastocerus* is related to *Anostirus*, i.e., the member of Dendrometrinae.

### 3.4. Taxonomy

#### Hapatesinae, New Subfamily

urn:lsid:zoobank.org:pub:349153F9-24A6-49F1-B595-6A6A76004A95.

Type-genus: *Hapatesus* Candèze, 1863.

*Differential diagnosis*. Neither the earlier placement in Dendrometrinae: Ctenicerini [54] (=Denticollinae *sensu* [1], =Athoinae *sensu* [55]) nor the herein proposed transfer to the clade of Hapatesinae, Parablacinae, and Pityobiinae are supported by reliable diagnostic morphological characters and our transfer is based on molecular analyses and the statistical tests of alternative relationships (Table 2). Conversely, the genera *Hapatesus* and *Toorongus* are morphologically very similar [54] and they share a combination of several unique characters which enable their reliable identification: a robust, basally constricted scapus, cylindrical pedicel and antennomere 3, flat laterally rounded antennomeres 4–10, spearhead-shaped antennomere 11, prosternum projected in a chin, pronotal lateral edge with two keels, elytra with rows of internal lens-like structures, each bearing a seta, male genitalia symmetrical (Figure 4 and Figure 5).

*Constituent genera*. *Hapatesus* Candèze, 1863 (21 spp.); *Toorongus* Neboiss, 1957 (4 spp.) [53,54,56].

*Distribution*. Confirmed records are known only from the Australian region [52]. All Australian records are known from the eastern mountainous regions, especially from southern Queensland, New South Wales, Victoria, and Tasmania; five species occur in New Guinea and New Britain [53,54,56].

Redescription of *Hapatesus* (Figure 4A–J, Figure 5A–P, and Figure 6K).

Body moderately convex, 5–10 mm long; greatest body width at base of pronotum, ratio length/width 3.06, whole body clothed with long setae (Figure 4A,B).

Head deeply inserted into pronotum (Figure 4C–E), behind eyes narrower (Figure 4F,G), frons deflexed, vertical. Eyes large, moderately protuberant, in resting position partly hidden in pronotum, their diameter about their minimum frontal distance. Antennal insertions in frontal position, covered by clypeus from above, widely separated (Figure 4D). Anterior edge of clypeus simple, with lateral bulges (Figure 4G). Mouth cavity anteriorly oriented. Cervical sclerites well-sclerotized. Antennae filiform to serrate, pubescent, with 11 antennomeres, reaching to middle of metasternum. Antennomere 1 constricted at base, robust, with posterior concave area fitting to surface of eye if antenna in resting position; antennomere 2 longer than antennomere 3, though both filiform; antennomeres 4–10 serrate with translucent, widely rounded lateral parts; antennomere 11 flat and spearhead-shaped (Figure 4B,I,J). Labrum partly visible, free, sclerotized, apex convex. Mandibles robust with subapical tooth. Maxilla with distinct galea and lacinia, mala setose; four maxillary palpomere, apical palpomere slender. Labium tiny, labial palpi 3-segmented, apical palpomere triangular.

Pronotum moderately convex with sinuate sides and projected posterior angles, widest posteriorly (Figure 4H). Prothorax with cavities for legs along prosternum. Lateral portion of prothorax with two parallel keels, inner keel reaching to two thirds of length of margin. Anterior angles of pronotum widely rounded, posterior angles moderately acute; posterior edge complex, fitting to corresponding parts of elytral humeri. Anterior portion of prosternum produced, widely rounded, chin hiding head from below (Figure 4H). Procoxal cavity circular, separated by basal part of prosternal process, posteriorly broadly open (Figure 4H). Clicking mechanism formed by long prosternal process and very deep mesosternal cavity reaching between mesocoxae (Figure 4H and Figure 5J). Scutellum well developed, abruptly elevated, anteriorly simple; posteriorly rounded (Figure 4F). Mesoventrite small (Figure 5J). Laterally with large spiracles (Figure 5I,K), mesocoxal cavities widely separated (Figure 5J). Metaventrite transverse, metaventral discrimen reaching to mid of length, without transverse groove (Figure 4J). Metacoxae narrowly separated, extended laterally, metacoxal plates well developed mesally, concealing most of femora.

Elytra with eight weakly impressed striae; seventh stria elevated at humeri and forms sharp edge, between striae rows of elytral punctures, each bearing seta, punctures covered by continuous surface of elytra and present as internal structures in elytron (Figure 5A–D,H). Wide epipleuron in humeral fourth, ending with two keels forming cavity (Figure 5B). Hind wing well developed (Figure 6K). Legs with apically widened, moderately long trochanters, femora robust, parallel-sided, tibiae slender (Figure 5E), outer edge of tibiae with setae, tarsi with five tarsomeres, claws slender, simple (Figure 5G).

Abdomen with five visible sternites, four basal sternites connate, terminal segments inserted in apex of abdomen, first ventrite not completely divided by metacoxae, with anterior process (Figure 5L), terminal segments as in Figure 5M,P. Aedeagus trilobate symmetrical, parameres enveloping phallus basally, outwardly hooked (Figure 5M–O). Female genitalia described earlier [53].

Eudicronychini Girard, 1971, new status.

Type-genus: *Eudicronychus* Méquignon, 1931 (=*Dicronychus* Laporte, 1840 *nec* Brullé, 1832).

Eudicronychinae Girard, 1971: 645 [57].

Eudicronychidae: Girard, [58,59,60] 1991, 2011, 2017.

Dicronychinae Schwarz, 1897: 11 (Type genus *Dicronychus* Laporte, 1840).

*Remark*. *Eudicronychus* and related genera are similar to other click beetles in most characters (Figure 6A–J,L). They were considered as a separate lineage with an uncertain position, sometimes outside the Elateridae system due to morphologically aberrant male genitalia (Figure 6H,I). The group has been given the rank of a subfamily in Elateridae, i.e., Dicronychinae [1,57,61] or a family, Dicronychidae [57,58]. The lineage includes four genera (*Eudicronychus* Méquignon, 1931, *Anisomerus* Schwarz, 1897, *Tarsalgus* Candèze, 1881, and *Coryssodactylus* Schwarz, 1897; [59,60]). The body is slender and slightly convex; the head is partly inserted in the pronotum; the clicking mechanism is well developed; elytra have longitudinal furrows and the hind wings have characteristic elateroid venation (Figure 6A–J,L). The type genus, *Eudicronychus*, is being characterized by closed prosternal sutures, antennae of 11 antennomeres, the last abdominal segment is not very convex, moderately sinuous on the sides; the first section of the metatarsi not swollen, but a little compressed on the sides (Figure 6J). The male genitalia are trilobate with asymmetrical phallobase, symmetrical parameres, and a short phallus that is usually enclosed by parameres. Dolin [62] pointed to similarity of their wing venation with Dicrepidiini of Elaterinae and Douglas [10] recovered *Eudicronychus* as a sister to *Elater* but in a distant position from other Elaterinae, including *Melanotus*. Based on the results of molecular analyses (Figure 3 and Appendix A), we lower the rank of Dicronychinae to Eudicronychini and place them in Elaterinae as a sister to Melanotini. Morphological data do not provide sufficient support for the herein proposed position. Constituent genera: *Eudicronychus* Méquignon, 1931 (~50 spp.), *Anisomerus* Schwarz, 1897 (11 spp.), *Tarsalgus* Candèze, 1881 (7 spp.), and *Coryssodactylus* Schwarz, 1897 (2 spp.; Girard 2011).

## 4. Discussion

### 4.1. Mitogenomics of Elateridae

Only some click beetle mitogenomes have been available for four major click beetle subfamilies (Elaterinae, Dendrometrinae, Cardiophorinae, and Agrypninae; [33,34,35]). We report eight newly assembled mitogenomes in the present publication (Table 1) and these represent further six subfamilies (Hapatesinae, Parablacinae, Pityobiinae, Plastocerinae, Morostomatinae, and Eudicronychinae which are herein lowered to a tribe of Elaterinae). The structure of additional mitogenomes (Figure 2A and Appendix A) confirms the conservative arrangement of genes and low length variability of click beetle genomes as reported earlier [33,34,35,36,37].

### 4.2. Phylogenetic Relationships of Click Beetle Subfamilies

Parablacinae, Pityobiinae, Hapatesinae, and Morostomatinae are deeply rooted click beetle lineages (Figure 3 and Appendix A). Except *Pityobius* which is only distributed in the Nearctic region, all genera placed in these subfamilies share a Gondwanan distribution: *Tibionema* (Pityobiinae) is Neotropical, *Hapatesus* and *Toorongus* are Australian, all eight genera of Parablacinae are Australian, and four genera of Morostomatinae are Afrotropical. Here, we found that three of these subfamilies form a monophylum (Hapatesinae (Pityobiinae, Parablacinae)) and that Morostomatinae is another deeply rooted lineage with a Gondwanan distribution (Figure 3). These results mean that Pityobiinae (here only represented by *Tibionema*) return to close relationships with Parablacinae genera as has been proposed by Calder [53] and rejected by Kundrata et al. [20]. As these groups are morphologically aberrant, we keep the latest classification scheme pending further work on their relationships. There are several subfamilies of click beetles for which are not available mitogenomes, and in some cases even any molecular data. These are Neotropical Campyloxeninae, European neotenic Omalisinae, and several small lineages as Physodactylinae, Subprotelaterinae, Hemiopinae, Oestodinae, Tetralobinae, Thylacosterninae, and Lissominae [2,13,14,15,16,20,21,22,23,24,25,26]. We considered the earlier recovered molecular relationships of the later five and they have never been recovered in relationships to our focal taxa *Hapatesus* (Hapatesinae) and *Eudicronychus* (Eudicronychini). Additionally, they are morphologically well defined and do not share morphological traits with our focal taxa (Figure 3, Figure 4 and Figure 5). Therefore, their absence in the present dataset does not question our results leading to a redefinition of supergeneric taxa (See Taxonomy section). No molecular data are available for two lineages, i.e., Physodactylinae and Subprotelaterinae. These are small morphologically unique subfamilies of click beetles which also substantially differ from *Hapatesus* and *Eudicronychus.* Unfortunately, their distant position from *Hapatesus* and *Eudicronychus* can be inferred only from morphological traits.

The morphological diagnostic characters are often unreliable for definitions of subfamilies in Elateridae and especially the characters which would support the inter-relationships among principal lineages are almost absent [1,10]. Therefore, click beetle genera are often transferred between subfamilies. Similar problems with an inference of relationships were encountered when Sanger data were analyzed. The molecular mutation rate is much slower in Elateridae than in most of their soft-bodied relatives [2,63] and resulting trees have regularly weakly supported backbone [2,13,14,15,16,20,36,63]. As a result, the root of click beetles can be recovered in variable positions. For example, Kundrata et al. [14] recovered Negastriinae and Cardiophorinae at the deepest split of click beetles and Elaterinae as a derived lineage [64], but Kundrata et al. [20] conversely presented Elaterinae as a sister to other subfamilies and Negastriinae and Cardiophorinae as terminal lineages. Now, transcriptomics robustly supports the second alternative [22,25,26], and we also recover such relationships with our dataset using mitochondrial genomes (Figure 2 and Appendix A).

The first mitogenomic analyses covering the main lineages of click beetles suggest some new relationships. First of all, we recovered *Hapatesus* [54], earlier placed in Denticollinae (currently Dendrometrinae; Calder 1998), as a sister to Pityobiinae and Parablacinae (Figure 2 and Appendix A). The position of *Hapatesus* was stable across analyses and all tests rejected the earlier position within Dendrometrinae, as a sister to Dendrometrinae or a sister to the DMCA clade (Table 2). Therefore, we propose a new subfamily Hapatesinae. Pityobiinae earlier recovered as a distant lineage (although without any statistical support; [20]) is robustly supported as a sister-lineage to Parablacinae [53]. Morostomatinae, earlier recovered as a serial split with Parablacinae [20] is another relatively species-poor, deeply rooted Gondwanan lineage. The group was earlier considered as related to Dendrometrinae: Senodoiini Schenkling, 1927 (=Protelaterini Schwarz, 1902), and Dolin [65] erected for *Morostoma* Candèze, 1879, and related genera a separate subfamily. Our results support the subfamily rank, but due to limited support we suggest that the inferred position needs further data to be robustly resolved and that Morostomatinae cannot be excluded as a candidate member of the clade of predominantly Gondwanan Hapatesinae, Pityobiinae, and Parablacinae. Similarly, more data are needed for the placement of *Plastocerus* (Plastocerinae), now recovered within Dendrometrinae in several analyses (Appendix A).

The earlier studies of the Elateridae consistently gave a high rank, either a subfamily or even a family to the eudicronychine lineage. We included in our analysis *Eudicronychus* spp. and recovered these representatives of Eudicronychinae as a terminal lineage in Elaterinae in a sister position to *Melanotus* sp. or a deeper position (Figure 3 and Appendix A) The support of such relationships is robust, and we prefer their placement in Elaterinae instead of earlier subfamily-rank classification which was mainly based on an aberrant morphology of male genitalia [57,58,59,65,66,67]. Therefore, the rank of this group is lowered to the tribe in Elaterinae.

We cannot say that the extent of the proposed changes is unexpected as all previous analyses have clearly indicated that Sanger data lack the power to reliably recover the relationships among major click beetle lineages and repeated analyses of the same dataset or its subsets produced conflicting results [13,14,20]. The more important question is why combinations of 18 rRNA, the D2 loop of 28S rRNA, *cox1* mtDNA, and *rrnL* mtDNA perform so poorly. One reason is a low mutation rate of all these fragments compared to those of closely related soft-bodied elateroids. When multiple alignments are produced for elateroid rRNA sequences, the stem regions are easily aligned but do not provide sufficient variability for phylogenetic inference and their loop regions are so length-variable that the alignment is unreliable. Additionally, we found that *cox1* and *rrnL* mtDNA fragments of click beetles are also highly similar and most mutations in the *cox1* gene are found in 3rd positions. We intentionally used as an outgroup only a single taxon for this analysis and used mitogenomes which mostly consist of protein-coding genes. In such a way, we avoided problematic alignments of length variable loop regions of 18S and 28S rRNA genes. Besides, we now have an opportunity to validate the inferred topology with earlier phylogenomic topologies which are based on >4000 orthologs [26]. Additionally, we encountered in our data strong base compositional and mutational rate heterogeneity which may violate the stationarity assumption of the widely used site-homogeneous models of nucleotide substitution [68,69] and may negatively affect the reconstruction of phylogenetic relationships. Possibly, the low variability and compositional heterogeneity caused an aberrant topology recovered by individual analyses. Recent phylogenetic studies using mitochondrial genomes demonstrated that the site-heterogeneous mixture model (CAT-GTR model) implemented in PhyloBayes tends to reduce tree reconstruction artifacts [34,70,71]. Our BI analyses used site heterogeneous models and the results show their power for resolving phylogenetic relationships with Elateridae and provide for an additional example that model adequacy is critical for accurate tree reconstruction in mitochondrial phylogenomics [34]. Multiple tree searches and thorough analyses aimed to test the effect of random trapping of the analysis in a local optimum [72]. In such a way, we produced the phylogeny with a much higher statistical support. We suggest that intensive sequencing of mitochondrial genes and their analyses compared, eventually constrained by a backbone based on phylogenomic data, can produce in a relatively short time and with feasible costs a robustly supported natural classification of click beetles.

## 5. Conclusions

We report mitochondrial genomes for an additional six subfamilies of click beetles (Eudicronychinae, Hapatesinae subfam. nov., Parablacinae, Pityobiinae, Plastocerinae, and Morostomatinae) and we infer their relationships with earlier sequenced Elaterinae, Dendrometrinae, Cardiophorinae, and Agrypninae. The phylogenetic analyses produced a fully resolved topology which suggests that *Hapatesus* is a sister to Pityobiinae and Parablacinae, and that *Eudicronychus* (earlier Eudicronychinae) is a member of Elaterinae. As a consequence, the new subfamily Hapatesinae is erected and Eudicronychini stat. nov. is transferred to Elaterinae. The mitogenomic phylogenetic topology generally agrees with results of earlier phylogenomic analyses which resolved the deepest relationships [21,25] and the mitochondrial genomes are sufficient for the resolution of shallower bipartitions.

## Figures and Tables

**Figure 1 insects-12-00017-f001:**
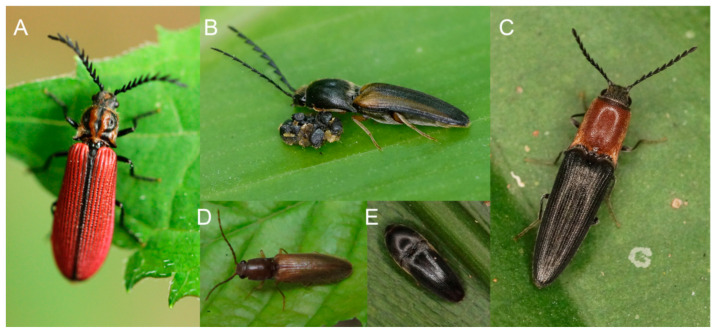
Clicking elateroids in nature. (**A**–**D**) click beetles, Elateridae; (**E**) false click beetle, Eucnemidae.

**Figure 2 insects-12-00017-f002:**
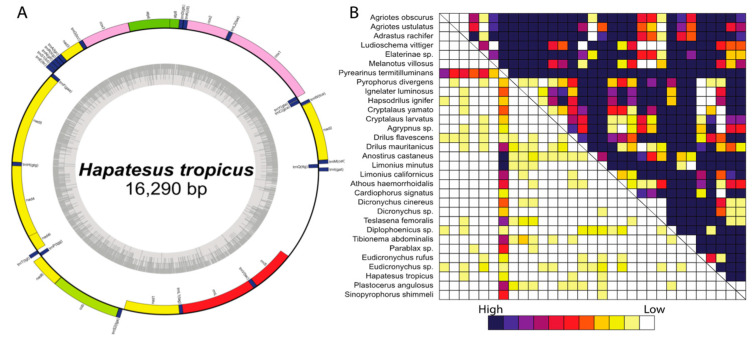
(**A**) A scheme of the circular mitogenome of *Hapatesus tropicus* Neboiss, 1958; (**B**) rectangular heat map calculated with SymTest showing *p*-values for the pairwise Bowker’s tests in the datasets C (amino acids, lower triangle) and A (nucleotides, upper triangle). Darker boxes indicate lower *p*-values and thus larger deviation from evolution under stationarity, reversibility, and homogeneity (SRH) conditions.

**Figure 3 insects-12-00017-f003:**
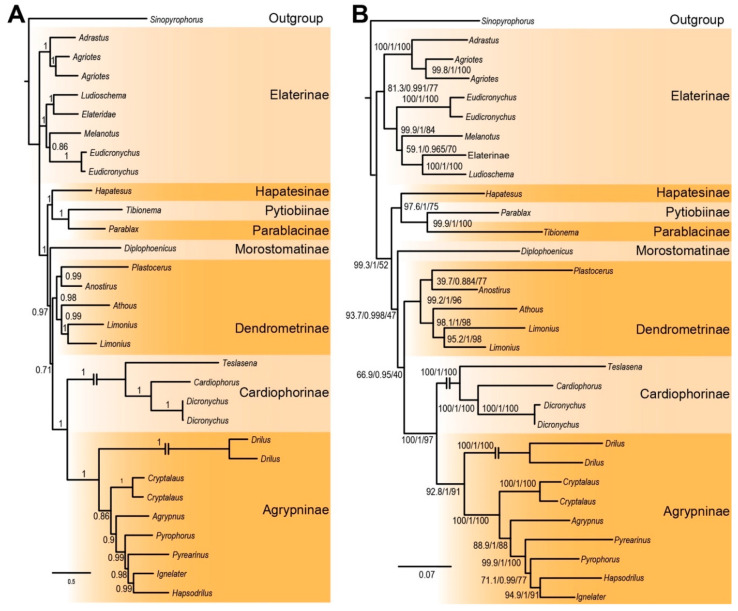
(**A**) Bayesian tree inferred from the unpartitioned dataset NUC123 in PhyloBayes under the site heterogeneous mixture CAT and GTR model. The values at nodes are Bayesian posterior probabilities; (**B**) Maximum likelihood tree from the IQ-TREE analysis of the dataset PCN12. The branch support values represent SH-aLRT, aBayes test, and ultrafast bootstrap.

**Figure 4 insects-12-00017-f004:**
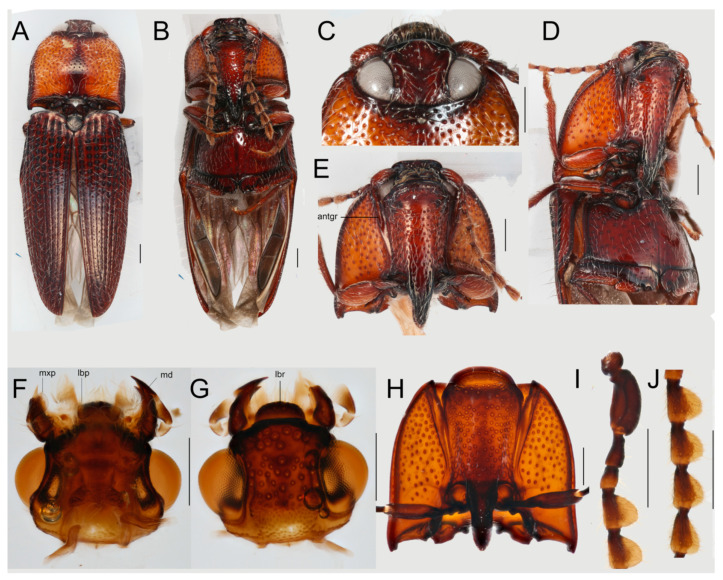
*Hapatesus tropicus* Neboiss, 1958. (**A**) General appearance, dorsal view; (**B**) ditto, ventral view; (**C**) head, dorsal view; (**D**) head and thorax, ventro-lateral view; (**E**) prothorax, ventral view; (**F**) head, ventral view; (**G**) head, dorsal view; (**H**) prothorax, ventral view; (**I**) basal antennomeres; (**J**) antennomeres 4–7. Scales 0.5 mm. Abbreviations: antgr—antennal groove, mxp—maxillary palpi, lbp—labial palpi, md—mandible.

**Figure 5 insects-12-00017-f005:**
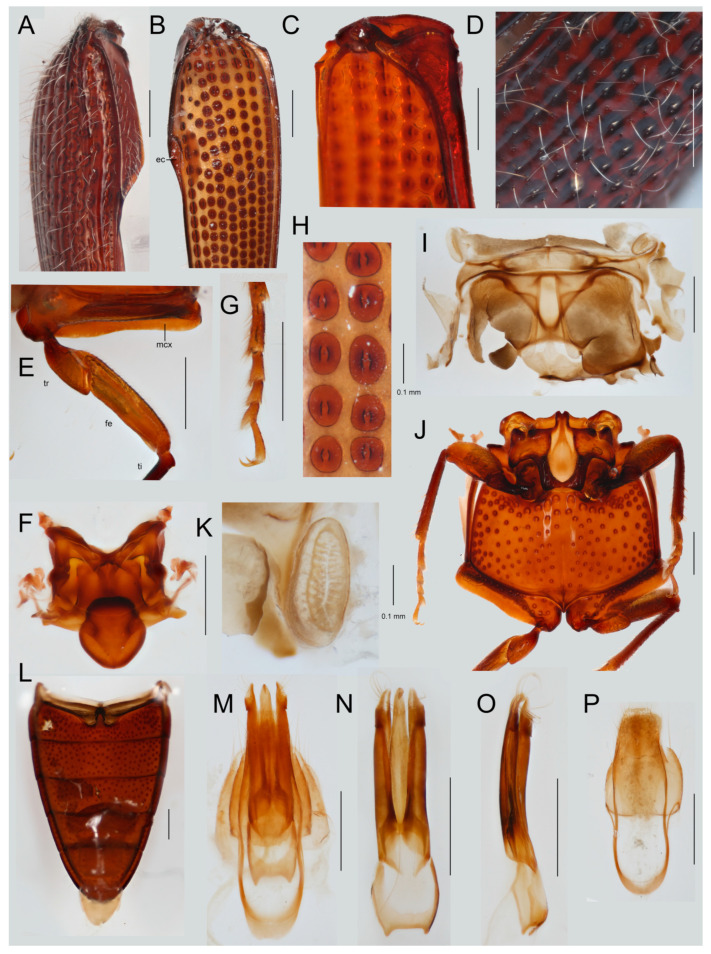
*Hapatesus tropicus* Neboiss, 1958. (**A**) right elytron, dorsal view; (**B**) ditto, ventral view; (**C**) elytral humeral part of the elytron, ventral view; (**D**) surface of elytron, detail; (**E**) metacoxa, trochanter, and femur; (**F**) mesoscutum, ventral view; (**G**) mesothoracic tarsus; (**H**) detail of elytron in ventral view; (**I**) metanotum, dorsal view; (**J**) meso- and metathoracic sternum ventral view; (**K**) mesothoracic spiracle; (**L**) abdominal sterna; (**M**) terminal abdominal segments and male genitalia; (**N**) male genitalia, ventral view; (**O**) male genitalia, lateral view; (**P**) terminal abdominal segments. Scales 0.5 mm if not designated otherwise. Abbreviations: ep—epipleural cavity, mxc—metacoxa, tr—trochanter, fe—femur, ti—tibia.

**Figure 6 insects-12-00017-f006:**
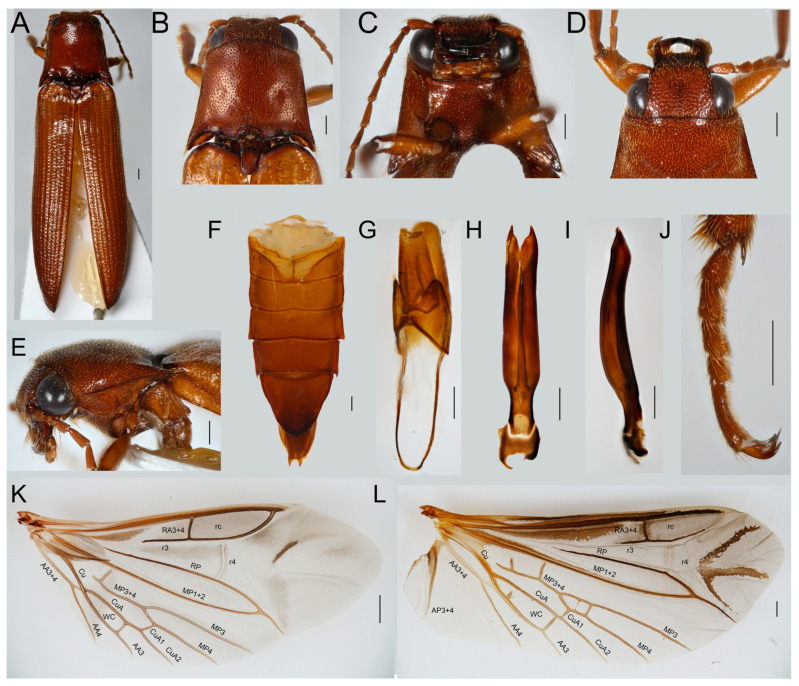
*Eudicronychus rufus* Fleutiaux, 1919. (**A**) General appearance, dorsal view; (**B**) head, prothorax and elytral humeri, dorsal view; (**C**) head and pronotum, ventral view; (**D**) head, dorsal view; (**E**) head, prothorax, and the base of elytra lateral view; (**F**) abdominal sterna; (**G**) terminal abdominal segments; (**H**) male genitalia, ventral view; (**I**) male genitalia, lateral view; (**J**) metatarsus; (**K**) metathoracic wing of *Hapatesus tropicus*; (**L**) metathoracic wing of *Dicronychus rufus* Scales 0.5 mm. The vein designation follows [1].

**Table 1 insects-12-00017-t001:** The sequenced samples of Elateridae included in this analysis.

Voucher Number	Genus, Species	Subfamily	Tribe	Geographic Origin
G18004	*Drilus mauritanicus*	Agrypninae	Drilini	Spain
G19011	*Diplophoenicus* sp.	Morostomatinae		Madagascar
G20012	*Tibionema abdominalis*	Pityobiinae		Chile
G19006	*Parablax* sp.	Parablacinae		Queensland
G19007	*Eudicronychus rufus*	Elaterinae	Eudicronychini	Zambia
G20004	*Eudicronychus* sp.	Elaterinae	Eudicronychini	Zambia
G20007	*Hapatesus tropicus*	Hapatesinae		New Guinea
A01544	*Plastocerus angulosus*	Dendrometrinae and/or Plastocerinae		Turkey

**Table 2 insects-12-00017-t002:** Results of likelihood test constrained trees versus an unconstrained tree. DMCA clade—a clade containing Dentrometrinae, Morostomatinae, Cardiophorinae, and Agrypninae; deltaL—logL difference from the maximal logl in the set; bp-RELL—bootstrap proportion using RELL method; p-KH—*p*-value of one-sided Kishino–Hasegawa); p-SH—*p*-value of Shimodaira–Hasegawa test; p-WKH—*p*-value of weighted KH test; p-WSH—*p*-value of weighted SH test; c-ELW—Expected Likelihood Weight; p-AU—*p*-value of approximately unbiased (AU) test.

Topology	logL	Deltal	bp-RELL	p-KH	p-SH	p-WKH	p-WSH	c-ELW	p-AU	Validity
Unconstrained position	−195,557.3	0	0.987	0.994	1	0.984	0.998	0.987	0.992	accepted
Sister to DMCA clade	−195,590.6	33.30	0.001	0.006	0.117	0.006	0.012	0.001	0.002	rejected
Sister to Dendrometrinae	−195,594.0	36.68	0.012	0.016	0.073	0.016	0.039	0.012	0.012	rejected
Terminal in Dendrometrinae	−195,726.3	169.0	0	0	0	0	0	2.83 × 10^−25^	2.84 × 10^−39^	rejected

## Data Availability

The newly produced mitogenomes are publicly available in GenBank under voucher numbers listed in Table 1 and the datasets are available in Mendeley data depository (doi:10.17632/gv88vzb7vn.1).

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
