# Peer review of "Click Beetle Mitogenomics with the Definition of a New Subfamily Hapatesinae from Australasia (Coleoptera: Elateridae)"

_insects, 2020, doi:10.3390/insects12010017_

Round 1

Reviewer 1 Report

line 1: I would delete „The“ in the title. Isn´t it general „Click beetle mitogenomics“?

line 11: there are no mitogenomes of subfamilies, only of individuals or species; better say „ mitochondrial genomes of X species belonging to six different subfamilies.“

Lines 13-14: Phylogenetic analyses suggest that Eudicronychinae is not a deeply rooted lineage in click beetles and represents a terminal lineage in Elaterinae.

line 14: delete „a“ in front of „new status“.

Line 18: „one of the families“ a taxon

Line 19-20: I would oppose morphology and molecular data and delete „ribosomal, and mitochondrial“ Differentiation of the latter is confusing here.

Lines 22-23: isn´t it possible to delete: „the constant pattern found in“ ?

Line 25: Insert article: for the Australian

Line 31: An Eudicronychine

Line 40: well-known with a hyphen

Line 40: specialists, plural!

Line 42: of „similarly clicking false click“

Line 76: Delete article „The“ Next-generation sequencing…

Line 91: „its internal structure“ this is confusing and appears to refer to morphology. It probably refers to classification of subfamilies and tribes, correct? Please rephrase.

Line 91: Herein, „With the material from recent expeditions,

Lines 101-102: kept till isolation at -20 °C until extraction of DNA,

Line 103: „ethyl alcohol“ better say „ethanol“

Line 124: Insert article; the MITOS2 webserver

Table 1: … in the this analysis.

Line 134: with an additional 23

Line 135: (22 ingroup taxa and 1 outgroup taxa)

Line 140: and 2 rRNA: spell out „two“

Line 174: insert article: and generated a

Line 174: insert article: we used the program

Line 182 ff.: was placed to (A) in a sister …. (B) in a sister…. (C) in a

Line 190: spell out „two“ rRNA

Line 191: of the newly sequencesd

Lines 193-94: Rephrase the two sentences. You can give the taxon names and the size of the mitogenome in one sentence.

Line 210: no verb: …was 0.95—0.97

Lines 215-216: „There were recovered two major clades“ Better say „Two major clades were recovered“

Line 219: „,was a“ delete „a“

Line 220: „relationships“ should be singular

Line 223: the relationships: what relationships is here referred to? Please clarify.

Line 224: Hapatesus should be in italics

Line 227: „was found in the relationships to three“ I do not understand this. Should it mean that they are in one clade?

Lines 230-231: „firmly held by the robust support for deep bipartitions“ shouldn´t it mean something like „is well-supported“ please rephrase

Line 233: „analyses congruently confrim“ better say „all analyses confirm“

several lines: please decide if you use „dataset“ as one word or „data set“ Both are used in several places.

Line 252: Hapatesinae, a new subfamily: should be without the article

Line 253: Hapatesus should be in italics

Line 297: I Havel never heard of the term „metathoraxic coxa“ Shoudn´t it be „metacoxa“??

Line 302: „posterior edge of complex“ delete „of“ ?

Line 305: „Promesothoracic clicking mechanism“ Is this a common term for click beetles? I would simply say „Thoracic clicking mechanism“. The combination of Pro and meso sounds very strange to me.

Line 308: „Spiracula“ I would say „spiracle“

Line 312: Elytra more than (?) eight weakly impressed striae;

Line 322: Delete „was“. If you insist keeping a verb, it should be „were“ as genitalia is plural.

Line 329: and t. They were   make a separate sentence

Line 335: „longitudinal striae“ Does this refer to the morphological term „striae“ or a descriptive one that should better be „furrows“ or something like that. Striae are always longitudinal.

Line 337: delete „is“ or use construction with „being“

Line 338: say „metatarsi“ instead of „tarsi of the hind legs“

Line 339: Remove line break

Line 341: „with Elaterinae: Dicrepidiini“ say: „with Dicrepidiini of Elaterinae“

Line 345: „The morphology“ Either delete article, or, say „Morphological data“

Line 352-3: „(J) metathoracic tarsus“ say „metatarsus“

Line 361: say „a tribe of Elaterinae“

Line 372: „return to the close relationships“ say: „return to a close relationship“

Line 382: „as sisters to another click beetles“ another click beetle subfamily? Not sure what it is supposed to mean; The same mistake is being repeated in Line 384. Copy-paste error?

Line 384. „transcriptomics“ without article

Line 388: delete „have been“

Line 392: delete „to establish“

Line 441: „for an additional“

Author Response

Reviewer 1

Answer.

We are very obliged for the long list of improvements of our manuscript and we fully accepted all recommendations. Therefore, we do not answer them point-by-point. Again, many thanks for your effort invested in the review, correcting our mistakes, and the considerable improvement of our manuscript.

Reviewer 2 Report

The phylogenetic relationship elaterids is a very interesting topic for coleopterists. In this manuscipt the authors were trying to place the genus Hapatesus from Australasia in Elateridae based on a mitogenome-based phylogenetic approach. The comprehensive phylogenomic phylogenes seem to be comprehensive, but the systematic position of Hapatesus remains unresolved to my understanding. 1. The authors failed to include all extant elaterid subfamilies in the phygenetic analyses. About five (or more) subfamilies were not included in the analyses, due to a lack of mitogenomic data. As such, the conclusions in the paper that Hapatesus is sister to Pyt+Para is not grounded. 2. As shown many previous studies, the site-heterogeneous mixture model fits the mitogenomic data better than the site-homogeneous models in ML framework. Even though the authors did extensive partition-based ML analyses; these results seem misleading and useless to me. The CAT-GTR trees are more informative, but as shown in the results (Fig S7) the deeper nodes are never consistent or resolved. In the PCN12 dataset, the support for Hapatesus and their sister group is 0.79, which means weakly supported. 3. The authors seemed to overlooked works by other colleagues. You should present the reason why not to cite the very relevant paper? A recent paper about the similar topic is not cited: Congruence Between Molecular Data and Morphology: Phylogenetic Position of Senodoniini (Coleoptera: Elateridae). Insects 2019, 10, 231; doi:10.3390/insects10080231. The authors did generate new data (new mitogenomes newly sequenced), but I am afraid I cannot recommend publication in the current form.

Reviewer 3 Report

Click beetles show a high morphological uniformity because of their clicking mechanism, which makes it difficult to use morphological data for taxonomic classification. Previously available molecular data have also provided only weak supportfor the click beetle classification.

Authors here report the mitochondrial genomes for additional six click beetle subfamilies, partly from the Austalian fauna (the Australian fauna is especially important because of their ancient diversity), and present a novel classification of the Elateridae. The most important results is a new subfamily of Hepatesinae and that Eudicronychini stat. nov. is transferred to the Elaterinae. The convincing molecular data are supported by some morphological redescriptionsR. 

The study is very well done and the manuscript is also written well. My congratulations to the authors for that work. I have only few suggestions for improvement:

  • line 70: delete the bracket after information
  • line 113: insert a space between number and unit
  • References 3, 43, 44, 48 and 49: use lowercase letters for the journal titles

Author Response

Reviewer 3

Click beetles show a high morphological uniformity because of their clicking mechanism, which makes it difficult to use morphological data for taxonomic classification. Previously available molecular data have also provided only weak support for the click beetle classification.

Authors here report the mitochondrial genomes for additional six click beetle subfamilies, partly from the Australian fauna (the Australian fauna is especially important because of their ancient diversity) and present a novel classification of the Elateridae. The most important results are a new subfamily of Hapatesinae and that Eudicronychini stat. nov. is transferred to the Elaterinae. The convincing molecular data are supported by some morphological redescriptions.

The study is very well done, and the manuscript is also written well. My congratulations to the authors for that work. I have only few suggestions for improvement:

line 70: delete the bracket after information

line 113: insert a space between number and unit

References 3, 43, 44, 48 and 49: use lowercase letters for the journal titles

Answer. All suggestions accepted and the text modified.

Round 2

Reviewer 2 Report

The authors did extensive revision of the MS and replied to my comments with caution and careful consideration (added discussion about the taxon sampling). I now feel easy to say that the revised MS is suitable for publication.